



**Coupling biophysical processes and water rights to simulate spatially distributed**
**water use in an intensively managed hydrologic system**
**Bangshuai Han[1], Shawn Benner[2], John Bolte[3], Kellie B. Vache[3], Alejandro N. Flores[2]**
*[1] Natural Resources and Environmental Management, Ball State University, Muncie, IN, 47306, USA*
*[2] Geosciences, Boise State University, Boise, ID, 83725, USA*
*[3] Biological & Ecological Engineering, Oregon State University, Corvallis, OR, 97333, USA*
*Correspondence to: Alejandro N. Flores (LejoFlores@boisestate.edu)*
**Abstract:** Humans have significantly altered the redistribution of water in intensively managed
hydrologic systems, shifting the spatiotemporal patterns of surface water. Evaluating water
availability requires integration of hydrologic processes and associated human influences. In this
study, we summarize the development and evaluation of an extensible hydrologic model that
explicitly integrates water rights to spatially distribute irrigation waters in a semi-arid agricultural
region in the Western United States, using the Envision integrated modeling platform. The model
captures both human and biophysical systems, particularly the diversion of water from the Boise
River, which is the main water source that supports irrigated agriculture in this region. In
agricultural areas, water demand is estimated as a function of crop type and local environmental
conditions. Surface water to meet crop demand is diverted from the stream reaches, constrained
by the amount of water available in the stream, the water rights-appropriated amount and the
priority dates associated with particular places of use. Results, measured by flow rates at gaged
stream and canal locations within the study area, suggest that the impacts of irrigation activities
on the magnitude and timing of flows through this intensively managed system are well captured.
The multi-year averaged diverted water from the Boise River matches observations well, reflecting
the appropriation of water according to the water rights database. Because of the spatially explicit
implementation of surface water diversion. The model can help diagnose places and times that
water resources is likely insufficient to meet agricultural water demands, and inform future water
management decisions.
**Highlights:**
• A novel tool that explicitly integrates water rights to spatially allocate irrigation
• Captures elements of both human and biophysical systems
• Inform future water management policies and decisions
**Keywords**: Integrated modeling; Treasure Valley; Irrigation; HBV; Water use; Water right



## 1    Introduction

### 1.1 Background

Increasing water demands for both agricultural and domestic consumption under the stress of climate change and increasing population represents a global environmental challenge [*Vörösmarty et al.*, 2000]. This increasingly limited hydrologic supply exists within the context of often extensive built hydrologic infrastructure. In turn, the management of that infrastructure is driven by complex social processes and decision making [*Pahl-Wostl*, 2007]. Accordingly, projecting how climate change and human activities will alter water availability in the future requires developing models that can integrate human decision making and biophysical processes [*Girard et al.*, 2015]. This challenge is particularly acute in arid and semi-arid regions where water resources are typically limited and actively managed to support irrigation-supported agriculture [*Falkenmark*, 2013].

Explicit integration of both human and environmental processes in hydrologic modeling is an area of active investigation and a variety of approaches are being used. For example, *Jakeman and Letcher* [2003] introduced attempts in Australia to integrate between hydrological and economic models using a nodal network approach. *Ahrends et al.* [2008] developed a coupled model system, consisting of a distributed hydrological model and an economic optimization model, communicating via model interfaces, to investigate regional interdependencies between irrigated agriculture and regional water balance in West Africa. *Ferguson and Maxwell* [2012] applied an integrated hydrologic model to compare effects of climate change and water management on terrestrial water and energy budgets of a representative agricultural watershed in the semi-arid Southern Great Plains of the United States. *Willaarts et al.* [2012] discussed win-win management solutions through societal evaluation of hydrological ecosystem services. *Cai et al.* [2013] evaluated potential hydrologic alterations of the Yangtze River under four scenarios of reservoir operation strategies by balancing human and environmental factors. *Kirby et al.* [2013] conducted a basin-wide simulation of flows and diversions for economic and policy analysis in the Murray-Darling Basin. *Laniak et al.* [2013] summarized recent progress and difficulties of integrated environmental modeling and urged that global community of stakeholders transcend social, and organizational boundaries and pursue greater levels of collaboration.

In the arid and semi-arid regions agriculture often relies heavily on irrigation and is typically the largest water use [*Döll and Siebert*, 2002; *Shiklomanov*, 2000]. Irrigation diverts water to the




originally dry lands, significantly altering the hydrological cycle. Because amount and timing of
applied irrigation water is, ultimately, a local decision made by farmers for individual fields, it is
particularly challenging to explicitly express these changes in a way that captures resulting
spatially and temporally variable impacts.
A variety of approaches have been taken to express irrigation in hydrologic models. Many models
rely on a simple soil-water balance module, and empirically estimate the agricultural water
demand. For example, *Gisser and Mercado* [1972] applied empirically estimated agricultural
water demand into a hydrologic model in Pecos Basi. *Döll and Siebert* [2002] developed a global
irrigation model to calculate the irrigation water requirements depending actual and potential
evapotranspiration rates. *Cai et al.* [2012] applied an irrigation diagnosis model to a regional
irrigation system in the Yangtze River Basin to analyze the local water budget. These models are
advantageous for hydrologic-economic assessment, but typically discount some details of the
physical system. Physically based models can simulate processes influencing the water balance,
including crop growth, irrigation, fertilizer applications and solute transport. Examples include the
soil-water-atmosphere-plant (SWAP) model [*Dam et al.*, 1997; *Droogers et al.*, 2000], the
Environmental Policy Integrated Climate (EPIC) model [*Gassman et al.*, 2005] and the CropSyst
model [*Stöckle et al.*, 2003]. Generally, these models are operated as point-scale models and do
not express processes in a spatially explicit manner. With expanded computational capacity and
the progress of GIS, increasing interest has been put on the integration of agricultural based
models with spatially-distributed hydrologic models, e.g., VIC-CropSyst [*Stöckle et al.*, 2014],
GEPIC [*Liu et al.*, 2007]. Generally, the commonly used approach in irrigation modeling is to set
soil moisture to field capacity or soil saturation or set a fixed evapotranspiration rate in irrigated
areas [*Leng et al.*, 2014], which can lead to inaccurate water budget and an inability to represent
irrigation in a more realistic way. It is also a common practice to assume unlimited water supply
when considering the sources and availability of irrigation water, which does not reflect the truth
in many water limited environments [*Sorooshian et al.*, 2012]. As such, we are aiming to
incorporate irrigation activities in our model in a more realistic way.
In the western United States, water is mostly allocated according to legally defined water rights
following the Prior Appropriation Doctrine, which basically defines that water rights are determined
by priority of beneficial use; historical use of water creates a right to the water. This means that
the irrigation amount is dependent not only on physically defined water availability, but also on
constraints dictated by legally defined water rights. In these systems water use for irrigation is,
therefore, the product of both environmental constraints (e.g. basin scale water availability and





evaporative demand) and human constraints through water rights allocations. Accordingly, water
rights represent an important, and well defined, constraint on irrigation water use in these
systems. However, few models take consideration of the influence of water rights on the
redistribution of water. The state of Texas has implemented a modeling system called Water
Rights Analysis Package (WRAP) to assess water availability and reliability of water resources
with local water rights [*Wurbs*, 2005a; b], but the model is not fully spatially distributed and the
model functions on a monthly scale.
In this study, we demonstrate an approach that integrates water diversion for irrigation based on
water rights within a physically-based model of hydrologic processes. We outline the development
of the core elements of both the biophysical and social system components of the model that
appear critical to represent the redistribution of water within the study area.
1.2 Study area
The Treasure Valley area, located in southwest Idaho, is the most populous region of Idaho and
contains its three largest cities, Boise, Nampa and Meridian (Figure 1), but is also home to an
extensive irrigation-supported agriculture. The area collectively comprises about 40% of state's
total population, with an area of 3323 km$^2$. Farm land occupies about 40% of the total
landscape, with an area of 1289 km$^2$, and relies heavily on irrigation through about 1700 km of
constructed canals.
Climate is generally semi-arid Mediterranean pattern with a hot dry summer and cold wet winter,
with strong spatial and temporal fluctuations in temperature and rainfall. Annual rainfall varies
substantially within the basin from ~ 700 mm in the northeast foothills to ~ 200 mm in the
southwest at the Lake Lowell, with a historical average of about 296 mm/yr at Boise Air
Terminal weather station. About 50% of the total precipitation occurs during the non-irrigation
season. Like many intensively managed landscapes in semiarid and mountainous regions of the
world, a series of reservoirs upstream of the Treasure Valley regulate and homogenize flows out
of the upper basin into the Boise River. The lower-most of these reservoirs, Lucky Peak, is
operated jointly by the US Army Corps of Engineers and the Bureau of Reclamation for
purposes of flood control and irrigation water supply. From Lucky Peak Reservoir, the Boise
River exits the mountains and flows about 103 km (64 miles) northwestward through the
Treasure Valley to its confluence with the Snake River. The Treasure Valley is bounded to the
north by the Boise foothills and to the south by the Snake River. A number of canals and
diversion dams have been built along the Boise River water course to allocate water resources.





Among the largest of these canals is the New York Canal that diverts water directly from the
Boise River about 1.6 km downstream of the Lucky Peak dam. During non-irrigation season, the
New York Canal carries a portion of the water to fill Lake Lowell, a reservoir within the Treasure
Valley area, for use during the irrigation season. During irrigation season, the New York Canal
carries a significant portion of the water from the Boise River and diverts it into distributary
canals within the agricultural areas of the Treasure Valley. With the benefit of irrigation,
population in the Treasure Valley has been growing rapidly and consistently since the 1870s.
Urban growth and increasing irrigation activities drive land use change and reallocation of water
resources.  Despite the importance of water resources and potential threats of water scarcity,
there have been limited integrative studies regarding water availability and scarcity in this area.
The Idaho Department of Water Resources (IDWR) conducted the Treasure Valley Hydrologic
Project starting in 1996, aiming to develop a better understanding of water resources in the
Treasure Valley and to evaluate changes in regional and local groundwater conditions.
Supported by this project, *Petrich [2004b]* characterized and simulated groundwater flow in the
Lower Boise River Basin, and analyzed the water budgets for the regional aquifer system based
on 1996 and 2000 calendar-year inflow and outflow estimates [*Petrich*, 2004b; *Urban and*
*Petrich*, 1996]. Local, state, cities and some federal agencies have also supported or conducted
a few water demand studies that characterized the local land use and the associated domestic,
commercial, municipal, and industrial water demands. However, most of these studies are
conducted at the conceptual level by estimating total water budgets. *Xu et al.* [2014] conducted
a hedonic analysis to estimate the response of agricultural land use to water supply information
under the Prior Appropriation Doctrine. Their results are informative at the scale of the entire
Treasure Valley but also lack spatiotemporally dynamic components that could be used to
reveal particular locations in space and periods in time where water demand and supply are out
of balance. This research seeks a practical integration of the spatiotemporal detail that is
available in the water rights database with the local spatiotemporal dynamics of surface water
hydrology. An important outcome of this study is an extensible modeling framework that can
serve as a foundational tool to capture and evaluate the complex interactions between the
social and biophysical systems related to water use in an integrated way.


Figure 1 Study Area: the Treasure Valley.

2      Methods



2.1 Envision platform and datasets
The model developed in this study is based on the Envision modeling tool, a spatially explicit
integrated simulation platform that can be used to integrate elements of biophysical and social
systems [*J P Bolte et al.*, 2007; *Inouye*, 2014]. Envision provides a geospatial software
framework to coordinate the interoperation of component models used to represent essential
processes and properties of the coupled social and biophysical systems being simulated.
Envision has been used in a variety of projects, e.g., to develop alterative future scenarios
under three growth management strategies for the Puget Sound Region in Washington, US [*J
Bolte and Vache*, 2010], construct a land use / land cover (LULC) agent based modeling for the
Motueka catchment, Australia [*Montes de Oca Munguia et al.*], evaluate potential impacts of
climate change on vegetation cover in the Willametter River Basin, Oregon, US [*Turner et al.*,
2015], and understand coupled natural and human systems on fire prone landscapes [*Barros et
al.,* 2015].
In Envision, the spatial domain is represented by a collection of polygons, called Integrated
Decision Units (IDUs). Each IDU polygon is associated to important geospatial attributes
characterizing both biophysical and social properties (e.g., elevation, soil type, land use,
population density, disturbance history, water right code, irrigation decision etc.). The IDU forms
the fundamental spatial unit for integrated decision-making in Envision. The process of creating
the IDU computational domain is somewhat ad hoc and iterative, but is meant to balance the
competing demands of fidelity to spatial heterogeneity and associated computational cost. The
IDU computational domain was constructed through a process that initially converted raster-
based LULC information into a polygon layer by grouping adjacent sets of pixels with similar
land-use/land-cover classes into polygons. Small polygons derived from a single LULC pixel
within a larger polygon of a different land-use/land-cover class (i.e., with an area of 900 m$^2$ or
less) were identified and deleted. The final constructed computational domain for the Treasure
Valley consists of 32,508 IDUs (polygons).
A variety of dataset is required to build the model (Table 1), among which, spatial heterogeneity
in the model is mainly reflected by three spatially explicit datasets: land cover, elevation, and
meteorological inputs. The land cover data is collected from the Nation Land Cover Dataset,
using the data of 2011. The elevation data is collected from the National Elevation Dataset with
a spatial resolution of 30 m. The climate dataset is a spatially and temporally complete, high-
resolution (4-km) gridded dataset of surface meteorological variables created by bias-correcting
daily and sub-daily mesoscale reanalysis and assimilated precipitation from the NLDAS-2 using





monthly temperature, precipitation and humidity from Parameter-elevation Regressions on
Independent Slopes Model [*Abatzoglou and Brown*, 2012]. The stream network is defined from
the NHDPlus V2 dataset, which represents stream networks as node-based line coverages.
Segments between nodes are considered to be stream reaches and each IDU is assigned a
stream reach for the purposes of simulating hydrologic routing. Artificial channels such as
irrigation canals and drains are explicitly represented. However, as discussed below, they are
functionally captured using the WaterMaster module, which simulates the allocated water based
on water rights.

Table 1 Datasets used in the model


2.2 Hydrologic processes
In this study, we employ the module called Flow with a slightly changed Hydrologiska Byråns
Vattenbalansavdelning (HBV) [*Bergström and Singh*, 1995; *Woodsmith Richard D. et al.*, 2007]
plugin to represent hydrologic processes. Human interventions include reservoir operations, and
agricultural irrigation which is simulated by another Flow plugin called WaterMaster. The primary
focus of the current paper is to develop a framework to incorporate human activities, mainly
irrigation, at the watershed scale, and provide solid basis for future integrated scenario
projections.
Within Envision, the HBV model is applied in a semi-distributed way to delineated Hydrologic
Response Units (HRUs) within the study domain affording the use of spatially distributed
datasets such as daily gridded meteorological inputs, land cover, and elevation information
[*Inouye*, 2014]. Within the model, HRUs are delineated by aggregating adjacent IDUs that are
associated with a common LULC and similar elevation, and 4456 HRUs are composed.
Hydrologic processes are simulated at the HRU-level, with fluxes being distributed uniformly to
the IDUs within the HRU.
Here, we briefly describe the slightly changed HBV model (Figure 2). A catchment in the model
is conceptualized as a series of linked reservoirs and is divided into 6 layers in this study:
snowpack, melt, irrigated soil, non-irrigated soil, upper groundwater and lower groundwater.
Runoff from the HRUs from different layers is then routed to streams using linear outflow
equations. The water balance equation in Flow (HBV) can be described as:





$$P - ET - Q = \frac{d}{dt}[SP + SM + UZ + LZ + lakes]$$
Eq. 1

where, $P$ = precipitation; $ET$ = evapotranspiration; $Q$ = runoff; $SP$ = snow storage; $SM$ = soil moisture
storage; $UZ$ = upper groundwater storage; $LZ$ = lower groundwater storage; $lakes$ = lake storage.
The model simulates daily discharge using daily rainfall, temperature, and potential
evapotranspiration as inputs. Precipitation is simulated to be either snow or rain depending on
whether the temperature is above or below a threshold temperature (TT). Rainfall and snowmelt
are then divided into water either filling the conceptual soil layer or recharge into groundwater
depending on the current soil moisture, field capacity (FC), and the parameter "Beta" (Eq. 2).
$$F = \left(\frac{Soil\ Water}{FC}\right)^{\beta}$$
Eq. 2

where, F is the fraction of rain or snow. Evapotranspiration (ET) is simulated using the FAO56
Penman-Monteith method as specified by the UN Food and Agriculture Organization (FAO) in
paper number 56 [*Allen et al.*, 1998] and in [*Allen and Robison*, 2007]. Generally, a crop
coefficient Kc is developed to simplify and standardize the calculation and estimation of crop
water use, and is an integration of the effects of crop properties and soil properties. As plants
grow and develop, Kc varies over time and the values are obtained from AgriMet Pacific
Northwest Cooperative Agricultural Weather Network. The potential ET of a specific crop, $ET_c$, is
then calculated as in Eq. 3:
$$ET_c = K_c * ET_r$$
Eq. 3

where, $ET_r$ is the reference evapotranspiration rate, the evapotranspiration rate for a
standardized vegetated surface corresponding to a living, agricultural crop (usually using full
cover alfalfa). For simplicity at this framework building stage, we do not include detailed crop
categories and crop rotation schedules. Rather, we use the crop coefficients of alfalfa for all
agricultural land use in the region due to the fact that most of the agricultural land in the
Treasure Valley is fully irrigated. Crop coefficients are assigned for non-agricultural lands based
on crop categories with a similar physical characteristics as an approximation (Table 2).
Detailed evapotranspiration calculation methods could be referred to [*Allen and Robison*, 2007].
Actual ET in the model is constrained by soil moisture at each HRU, as simulated in each daily
time step. The soil box is subdivided into two layers/fractions, irrigated soil and non-irrigated




soil, to help facilitate water to be irrigated and evaporated from the irrigation areas. The
response function consisting of two or three linear outflow equations depending on whether or
not recharge in the upper groundwater box (SUZ) is above a threshold value (UZL) then
transforms excess water from the soil layer to runoff (Eq. 4, Eq. 5, and Eq. 6).
$$Q0 = K0 \cdot (SUZ - UZL) \qquad \text{Eq. 4}$$
$$Q1 = K1 \cdot SUZ \qquad \text{Eq. 5}$$
$$Q2 = K2 \cdot SLZ \qquad \text{Eq. 6}$$
where, SUZ is the recharge (water depth) at the upper groundwater zone that is simulated at
each time step, UZL is a threshold value, SLZ is the recharge (water depth) at the lower
groundwater zone that is simulated at each time step. If SUZ >= UZL, then the total water that
is routed to runoff is the summation of Q0, Q1 and Q2. If SUZ < UZL, then the total water that is
routed to runoff is the summation of Q1 and Q2.
Table 2 Crop categories used to approximate the land use categories in the ET calculation
Figure 2 Flowchart of the Flow module in Envision. Note the human activities influencing water
availability. Water is distributed by the local water rights data (irrigation activities), and is also
constrained by the reservoir operations.
2.3 Simulation of water rights
Irrigated water allocation is simulated via a module called Watermaster (Figure 3) that adheres
to publicly available water rights data in Idaho in accordance with the Prior Appropriation
Doctrine [*Hutchins*, 1977; *Xu et al.*, 2014]. In this study, surface water and groundwater
irrigation activities are simulated based on the water rights data updated in 2012 by IDWR. Each
water right is associated with four attributes that are of critical importance to this study: (1) the
Place of Use (POU), (2) the Point of Diversion (POD), (3) the priority date, and (4) the
appropriated diversion rate.
The POU data is used to identify IDUs in the study domain with surface water and/or ground
water rights. For surface water rights, water is extracted from the stream reach closest to the
POD associated with that water right. In most cases in the Treasure Valley, the PODs are
located along irrigation canals not explicitly being simulated, and the PODs are assumed to be





the point at which water is originally diverted from a natural watercourse (A majority originally
diverted from the Boise River due to its seniority and largest diversion capacity) into the
associated supply canal system. The priority date of each water right determines whether or not
water can be diverted from the stream reach associated with the POD and applied to the IDUs
within a POU as irrigation on a particular date during the simulation. On each day of the
simulation, WaterMaster determines all water rights active on that date and, based on the
allocation rates of those water rights, determines the maximum flow of water that may be
diverted at each stream reach associated with one or more PODs. The irrigation water demand
at the POU is computed as the potential evapotranspiration for the agricultural IDUs within each
POU with a composite loss coefficient which is currently set based on an overall estimation of
60% water loss from the original diversion to ultimate crop use. The coefficient was roughly
estimated based on a local study of irrigation management in 1999 and the proposed potential
improvement in the study to reflect the current irrigation efficiency [*Huter et al.*, 1999]. The
amount of water demanded for diversion at the stream reach is then computed as the sum of
water demand for all POUs associated with a POD along that reach. If there is sufficient
streamflow to satisfy demand, the amount of water diverted equals the total demand. If there is
insufficient streamflow in the reach to satisfy demand, then water rights must be curtailed. Water
rights with highest seniority (i.e., earliest priority date) are satisfied and streamflow reduced by
the allocation rate associated with that right, followed by the next most senior water right, and so
forth until there is insufficient streamflow to meet demands of water right. At this point, that
water right and all more junior rights are curtailed only for the current date and will resume water
use whenever there is abundant stream flow later of the year. This approach simulates the
effect of canals and distributaries without explicitly simulating the hydraulics of canal flow.
Specifically, water is diverted from an actual place of diversion as captured by the IDWR
database and applied to a place of use in accordance with the water rights database. For
ground water rights, we assume unlimited groundwater source as of now due to the fact that
groundwater resources are abundant for the withdrawal rates in the Treasure Valley [*Petrich*,
2004a]. On the valley-wide basis, the volume of ground water pumped during the year accounts
only 15 to 20% of the total ground water recharge [*Urban and Petrich*, 1996]. Groundwater in
the Treasure Valley is mainly recharged from the seepages from the canal system, flood
irrigation and precipitation. Use of groundwater for irrigation is common, although surface water
rights comprise a much larger proportion of agricultural water use on a volume basis in the
Treasure Valley.
Here we would like to define a couple important terms used below:



The allocated water indicates the amount of water that is met and diverted to the corresponding
place of use in the model.
The unsatisfied water indicates the amount of water that is not met for the corresponding place
of use in the model.
The appropriated diversion rate is calculated based only on the POD rates and the
corresponding POUs, and reflects the amount of water that is potentially usable based on the
existing water right maximum rates while ignores priority dates and physical constraints. It is
calculated upon the water right dataset instead of being simulated by the model.

Figure 3 WaterMaster loop that makes use of the local water rights data for irrigation
2.4 Reservoirs and boundary condition
Reservoirs are considered part of the stream network in Envision. The location and physical
constraints of the Lucky Peak Reservoir and Lake Lowell's dams are set up based on the data
collected from the Hydromet database (Table 1). The Lucky Peak Reservoir receives water
drained from the watersheds upstream of Boise River, and is the main water resources for the
Treasure Valley. The historical inflows to the Lucky Peak Reservoir are used as inflow boundary
condition for the model. Lake Lowell is an offstream reservoir formed by three earthfill dams
enclosing a natural depression at southwest Treasure Valley. The reservoir naturally drains
water and is also filled during the non-irrigation season by diversions at the Boise River
Diversion Dam through New York Canal. In this study, we simplify the reservoir operations by
setting the maximum and minimum flows at a downstream control point of each reservoir (Boise
River at Diversion Dam for Lucky Peak Reservoir and Boise River near Parma River for Lake
Lowell) based on historical daily extreme values to regulate the extreme flow released from the
reservoirs. This setup is efficient while still simulates the normal operation of the Boise Project
Board of Control. The operation basically aims to control flood in the Boise River for the safety
of the city, uses the natural river flows until the Boise River falls to a certain level, and then
switches to water stored in reservoirs and provides users a certain allotment of water they can
use for the irrigation season. As such, by setting up maximum and minimum daily flows, the
reservoirs are designed to release water in the dry seasons and control flooding water in the
snow melt season of the area.
2.5 Model calibration and validation methods





The reliability of many hydrological models is dependent on calibration, which is the process of
finding an optimal set of parameters that enable the model to closely match the behavior of the
real system it represents [*Gupta et al.*, 1998]. We calibrated the model based on the Nash-Sutcliffe
coefficient (Eq. 7) between the observed and simulated stream flows at two USGS gages – Boise
River at Glenwood and Boise River near Parma, Idaho.
The Nash-Sutcliffe coefficient is calculated as:
$$E = 1 - \frac{\sum_{t=1}^{T}(Q_{obs}^{t} - Q_{sim}^{t})^2}{\sum_{t=1}^{T}(Q_{obs}^{t} - \overline{Q_{obs}})^2} \qquad \text{Eq. 7}$$
where, $Q_{obs}$ is the observed discharge; $Q_{sim}$ is the simulated discharge, and t is the time step at
calculation, $\overline{Q_{obs}}$ is the mean observed discharge over the entire run. Nash-Sutcliffe efficiencies
can range from - ∞ to 1 (perfect match). An efficiency of negative value indicates that the mean
value of the historical observations would be a better predictor than the hydrologic model.
Most parameters used in the model are estimated using a Monte Carlo approach. The data from
years of 2006 - 2009 are used for calibration processes, and from 2010 - 2013 are used for
validation purpose. For each run, each parameter value was randomly selected from a uniform
distribution; the minimum and maximum values of these distributions, listed in table 3, are
generally adopted from *Sælthun* [1996], *Lawrence et al.*[*Lawrence et al.*, 2009] and *Abebe et*
*al.*[*Abebe et al.*, 2010]. We simultaneously vary the values of the parameters within their target
ranges, and run the model 1000 times. Then the best-fit parameter sets are selected through an
assessment of the fit of simulated to observed runoff data based on visual inspection of fit and
Nash-Sutcliffe Efficiency (E) between the observed discharge and the simulated discharge. The
parameters are conceptually based on physical parameters of the system. Although they are
actually effective parameters that fit the model through calibration and do not necessarily
represent actual physical properties, it would be beneficial to get physically representative
values whenever possible. In this calibration process, we calibrate 9 parameters of the total
14 parameters, while setting 5 parameters constant to save computational time. The FC and
WP values were adopted from the SSURGO dataset from the Natural Resources Conservation
Service. Since LP, CFR and CWH are not sensitive to model performance [Seibert, 1997], a
reasonable LP value was set based on local soil conditions, and CFR and CWH were held
constant.
Table 3 Parameters used, the range considered for calibration and the calibrated values





3      Results
In this section, the calibration and validation results of the hydrological module are presented, the
water right dataset is summarized, and the irrigation water use and water scarcity from 2006 –
2013 are analyzed.
3.1 Calibration and validation
The model was calibrated and validated against historical observations through discharge at two
USGS gaging stations (Boise River at Glenwood and Boise River near Parma) and at the New
York Canal. These two calibration targets reflect influences of different processes. The upper
gaging station (Glenwood) is just down-stream from the New York Canal, the primary point of
extraction but is up-stream of the majority of return flow to the Boise River, which is primarily in
the lower portion of the river. In contrast, the Parma gaging station is located just above the
confluence with the Snake River and is downstream of both the majority of the extraction and
return flows. Accordingly, model results that successfully match the Glenwood gage provide a
good indication of the model's capacity to simulate water consumption and associated removal,
while comparing the model results to the Parma gage is more strongly influenced by the model's
capacity to capture return flow.

A plot of the simulated and the observed flows at these two USGS sites for the calibration

period (2006 – 2009) and the validation period (2010 – 2013) is shown in Figure 4. The model
effectively captures the major high and low flow events, the extreme values of which are
constrained by the downstream control points. For example, at Glenwood, the annual discharge
is clearly dominated by three periods associated with late winter or spring high flows, irrigation
season flows, and fall-winter low flows. The NS coefficient, which is a criterion to estimate the
goodness of fit between observational data and simulated data, is 0.82 during the calibration
period and 0.67 during the validation period at the Glenwood site, and 0.69 during the
calibration period and 0.62 during the validation period at the Parma site. The good fit to the
Parma gage suggests the model captures return flow particularly well. We also compare the
amount of water diverted to the New York Canal with the simulated results, and find a good
match with a correlation coefficient of 0.92 (Figure 5), indicating that the model does a good job
of capturing the diversion amount from the Boise River.



Figure 4 Simulated discharge and the observations during the calibration (2006 ~ 2009) and
validation periods (2010 ~ 2013) at the Glenwood Station of Boise River (Upper Panel) and
Parma Station of Boise River (Lower Panel).

Figure 5 Simulated irrigation amount and the observations averaged over the years of 2006 ~
2013 at the New York Canal. Blue color lines are daily discharge rate in m^3/s, and red color
lines are cumulative discharge in m^3.

3.2 A summary of the irrigation water rights
In the Treasure Valley, surface water is the main water source for irrigation, despite many more
POD's for groundwater. Currently, there are 22,217 PODs and 21,492 places of use (POUs) in
the study area, among which, 4,838 PODs and 3,859 POUs are appropriated for the irrigation use
(Figure 6). In the following analysis, all water rights are irrigation water rights unless stated
otherwise. Within all water rights database, 78% of the PODs use groundwater as water source,
and only 22% use surface water as water source. However, surface water is still the main water
source with regard to the amount of irrigated water supply. Surface water PODs are mainly located
along the Boise River, usually with a relatively higher maximum allowed diversion rate per POD
(maximum 38.21 m^3/s), while groundwater PODs are dispersed all over the irrigated lands,
usually with a relatively smaller maximum allowed diversion rate per POD (maximum 2.47 m^3/s).
Among all the surface water PODs, most surface water is mainly diverted from the Diversion Dam
which connects New York Canal with Boise River. Multiple PODs overlap at the Diversion Dam
with highly senior water rights, diverting about half of the stream flow from main branch of Boise
River during the irrigation season. The diverted water provides the water resources for Lake
Lowell and numerous irrigation canals downstream.
Figure 6 The spatial distribution of the Points of Diversion (PODs) for irrigation purpose, and the

maximum allowed diversion rates.

3.3 Model simulated spatial and temporal distribution of water use
Comparing simulated water use with that predicted based on appropriated rates suggests the
model does a good job of spatially distributing water use. The summarized appropriation rate
generally matches the boundary of the irrigation districts (Figure 7). According to the appropriation




rate, most of the water should be appropriated to the southwest part of the Treasure Valley, e.g.
Nampa-Meridian, and New York irrigation districts. In contrast, a relatively small amount of water
should be appropriated to those areas along the Boise River and into the Black Canyon irrigation
district which is located at the northwest part of the Treasure Valley.
Figure 7 The annual appropriated diversion rates calculated based on water rights maximum
allowed diversion rates and place of use, indicating the potential usable water. The irrigation
district boundaries and the names of major irrigation districts are also shown.
The model simulated allocation rate follows these spatial patterns of the appropriated rate (Figure
8). The southwest part of the study domain receives the most allocated water, while the northwest
part and the downstream section of Boise River is allocated less water (Figure 8).
Figure 8 The spatial distribution of the annual allocated irrigation water averaged over the
simulation period. The domain that is circled is Black Canyon Irrigation District, which receives
additional irrigation water from outside of the domain, where the water allocation is
underestimated.
The simulated water allocation confirms that surface water is the main water source with regard
to the amount of allocated water, as shown by the model simulated annual and monthly allocated
surface water rates, and allocated groundwater rates (Figure 9, Figure 10). The allocated surface
water discharge rate is ~ 21.3 m^3/s averaged over 2006 to 2013, while the allocated groundwater
is only ~ 4.0 m^3/s.
Figure 9: Average daily allocated surface water, groundwater and unsatisfied surface water use
for each year.
Figure 10: Average daily allocated surface water, groundwater and unsatisfied surface water use
for each month from 2006 to 2013.
The simulated water allocation also reflects the seasonal irrigation water use pattern. The
irrigation season in the Treasure Valley occurs from April to November when precipitation is rare
and temperature is high. As expected, most of the irrigation activities happens from May to
October, representing over 95.6% of the annual total irrigation amount. The peak irrigation season
is June, July and August, which irrigates 61.1% of the annual irrigation amount.
4      Discussions
4.1 The model's contribution to inform decision-making




### 4.1.1 The model reveals water scarcity and its causes by unsatisfied water distribution

The irrigation water scarcity is divided into 4 categories based on the annual unsatisfied irrigation water amount: Adequate Water Rights (< 100 mm deficit), Light Scarcity (100 – 300 mm deficit), Medium Scarcity (300 mm – 600 mm deficit), and Heavy Scarcity (> 600 mm deficit). There is less allocated water along the downstream section of Boise River, which also leads to higher water scarcity in the area (Figure 11). The northwest part of the study area experiences light to middle level water scarcity. Water scarcity is overall not serious in the Treasure Valley, however, could pose a problem in the relatively dry years such as 2007, 2008 and 2013.

Figure 11: The spatial distribution of the annual unsatisfied irrigation maps averaged over the simulation period. The domain that is circled is Black Canyon Irrigation District, which receives additional irrigation water from outside of the domain, where the water scarcity is overestimated.

On average, ~ 80.1% irrigation demand could be satisfied from 2006 to 2013, with an unsatisfied irrigation rate about 5.1 m^3/s for the whole irrigation area. However, the unsatisfied irrigation amount varies greatly between years. For example, in 2011 when the annual precipitation is higher than normal (Figure 12), only an annual average of 3.4 m^3/s irrigation amount is unsatisfied in the Treasure Valley, while in 2013 when the annual precipitation is lower than normal, the annual averaged unsatisfied irrigation amount doubled to about 5.9 m^3/s (Figure 9). The Mediterranean climate pattern produces dry-hot summers which, even in the wettest years, some degree of unmet water potential irrigation use.

Figure 12 Annual precipitation amount calculated at Boise Air Terminal (Station ID: 7268104131). Precipitation is calculated based on water year since irrigation in each calendar year is mainly affected by the precipitation during the spring and last winter.

While the water rights appropriation rate reflects the irrigation district regulation, the allocated rate also considers the biophysical demand, and has the capacity to reveal where the current water rights are not sufficient for biophysical use. For example, the areas along downstream Boise River experience a relatively higher water scarcity (Figure 11). Since the Boise River has abundant water to extract during the irrigation season as shown in the discharge figures (Figure 4), the water scarcity is mainly due to the water right constraints. While this area is ascribed to be agricultural land, the area is mainly used for grass/pasture (Figure 13), which does not require much irrigation. Should these areas be converted to irrigated agricultural lands, they will need a


larger water right allocation to support crops. This illustrates the value of spatially explicit demand-
based water allocation and associated patterns to understand the irrigation water use dynamics.
Figure 13 The spatial distribution of crops and grass/pasture in the agricultural area of the
Treasure Valley.


4.1.2 The model indicates irrigation inefficiency through the simulation of demand-based water
allocation and the actual water use
Demand-based water allocation rates and the actual water use vary significantly. The allocated
water in an IDU is determined by the IDU water demand, the water availability in the stream and
water rights allocation rate and priority. The IDU water demand is calculated for irrigated lands
based on the potential ET rates and the water loss coefficient. However, the actual water use by
the farmers is usually more arbitrary relying on their experience, their irrigation methods and the
economic expectations, and is a complex function. Application efficiencies for traditional furrow-
irrigated systems supplied by siphon tubes or gated pipe range between 30 - 40%, with
efficiencies of 50 - 60% percent possible with excellent management [*Neibling*, 1997]. A large
amount of water is wasted even in this water-limited environment. The simulated multi-year
average of allocated surface water is ~ 2.0 acre-feet per acre. This number is in the lower range
of the allotted irrigation water by the Boise Project Board of Control which is about 2 – 3 acre-
feet per acre in normal years for farmer use. This can also be validated by the diverted amount
of water from the New York Canal (Figure 5), with an overall slightly underestimation but very
good match between simulations and observations (correlation coefficient of 0.92). Considering
that the water release at the operational level normally relaxes the biophysical demands and
varies annually, our simulated irrigation water amount is in the right scale.
4.2 Model Limitations
While the model appears to be an effective tool to express spatially explicit water rights based
allocation, there are some important features not captured by the model. Specifically, during the
dry years, e.g., 2007 and 2013, the model produces higher simulated discharge compared to the
observations at the Parma River gage during the irrigation season. There are a number of reasons
for these deviations in the model: (1) Groundwater use is currently assumed to be unlimited,
leading to extra amount of water recharged into soil layer. Although this reflects the current





groundwater abundance of the study area, it does not maintain the water balance after
groundwater irrigation, and may lead to larger simulated stream discharge at the downstream of
the irrigation area. However, since groundwater irrigation counts for a very small portion of the
irrigation water use, we intend to simplify the model at this stage by assuming an unlimited
groundwater supply. (2) The diversion of water in many canals are actually operated as constant
flows, differing from the demand-need diversion rates of the model. As such, it is implausible to
find a perfect match between observations and simulations. (3) The model is limited to the Boise
River watershed and only water within that watershed is considered. However, there is some
transfer into the basin from the adjacent watershed. This is especially important for the northwest
part of the Treasure Valley (mainly Black Canyon Irrigation District) where the model predicts
water scarcity (Figure 11). In reality, some water is pumped from Payette River outside of the
boundary to irrigate this area so it is very likely that the model is underestimating the water
allocation and exaggerating the water scarcity in this area. (4) The model is semi-conceptual, and
ignores some minor consumptive water use. For example, the water that is incorporated into
products or crops, consumed by humans or livestock, or otherwise removed from the immediate
water environment.
A second area where the model underperforms is capturing some flow details at the beginning of
each year. Local agencies tend to empty the reservoirs in the winter time for spring flood
protection, while the model ignores this local human operation. In addition, irrigation water use is
not only affected by weather conditions and irrigation at the current time step, but also affected
by a longer term climate and surrounding environments. Considering that the surface water
source is mainly from snow melt in the upper Boise River Basin, the available water of an irrigation
season in the study area is strongly affected by the precipitation from the current spring and the
previous winter in the upper watershed. As such, the annual summation of the allocated water is
a complex nonlinear issue as shown in Figure 9. For example, 2007 is a dry year, but the allocated
water is still relatively high due to the high precipitation rate in 2006 which releases abundant
snow melting from the upper watershed during the earlier irrigation season of 2007.
Nonetheless, Boise River has never been totally drained out in a single day during the simulation
period, and has abundant water to be diverted. More accurate discharge matching the historical
record is not a deciding factor for irrigation activities in the area, and the downstream water
balance mismatch is currently not an influencing factor for irrigation distribution.



Despite the limitations and challenges, the results generated by this research have successfully
integrated irrigation activities into a hydrological model and can serve as a good start for further
studies. The current study also proves that the integrated modeling work can provide sufficient
spatial and temporal details to nevertheless provide useful insights into possible management
strategies for water use in the Treasure Valley.
4.3 Insights and future work
This work is built under a larger on-going modeling framework that aims to integrate complex
social and biophysical processes and reveals the requirement of multi-disciplinary corporation.
Our experience suggests that deploying such an interdisciplinary approach is by no means a
trivial task. During our research, a large team of scientists, engineers and stakeholders
continuously discuss and construct an agreement on the study domain which reflects both the
watershed boundary and political boundary, the research questions, the temporal scales and
the complexity of the work. Knowledge from local stakeholders are also borrowed to help justify
the design of the model. This research effort is an important step forward towards the solution to
the cultural and historical barrier to the integration across disciplines [*Hamilton et al.*, 2015].
As the first report of the modeling fruit, in this paper, we are using historical downscaled climate
data here to represent the climate, and the parameter set is only suitable for this specific case.
For the future research of water availability projection, a suite of different climate change
scenarios will be incorporated. Future modeling of this method will highlight changes in water
deficits over time by dynamically simulating IDU water demand and water availability. Water
rights are also going to be dynamically allocated with adoptive strategies when water scarcity is
more severe. In addition, other important factors such as urban growth, land use and land cover
change, and crop choice will also be integrated into the future model with the feedback of
stakeholders.
5. Conclusion
This study integrates spatially and temporally explicit irrigation activities into hydrologic cycles,
connecting agriculture, water rights and hydrologic processes in the semi-arid Treasure Valley.
The model results reveal the spatial and temporal patterns of irrigation water use, and areas
where current water rights are not always able to support irrigation demand. The model is useful
in that it can be used to diagnose places of use and times where allocated water is likely
insufficient to meet agricultural water demands, and inform future water management decisions.





The modeling framework is extensible and allows not only for the model to be subjected to future
scenarios of urbanization and climate change, but also as a tool for evaluating alternative future
scenarios of water management policies and actions. The model also indicates the current
knowledge gap in water use between the water rights based diversion rate and the actual irrigation
water consumption, including the complexity of human activities and the inability to fully capture
the discharge over dry years.
**Author contributions**: Bangshuai Han and Alejandro N. Flores designed this research and interpreted
the results. John Bolte and Kellie B. Vache provided technical support with debugging help. Bangshuai Han
prepared the manuscript with the help with Shawn Benner, Alejandro N. Flores, and get agreement for
submission with all co-authors.
**ACKNOWLEDGMENTS**
This publication was made possible by the NSF Idaho EPSCoR Program and by the National
Science Foundation under award number IIA-1301792. We appreciate James Sulzman and
Cynthia Schwartz for the help during the modeling processes.

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



Table 1 Datasets used in the model

| Input Data | Data Sources | Dates | Used in Model Components | Url |
|---|---|---|---|---|
| Land use/land cover | National Landcover dataset (NLCD) | 2011 | Evapotranspiration | http://www.mrlc.gov/nlcd2011.php |
| Streams/canals /Water bodies | NHDPlus V2 | 2012 | Building stream network and flow routing | http://www.horizon-systems.com/nhdplus/NHDPlusV2_17.php |
| Downscaled climate data | U of Idaho METDATA (4 km resolution) | 2006-2013 | Evapotranspiration | http://cida.usgs.gov/thredds/catalog.html?dataset=cida.usgs.gov/thredds/UofIMETDATA |
| Daily stream discharge | USGS Instantaneous Data Archive | 2006-2013 | Hydrology model calibration and validation | http://nwis.waterdata.usgs.gov/nwis/rt |
| Digital elevation model | NED (30 m resolution) | N/A | Building HRU | http://nationalmap.gov/elevation.html |
| Water rights | Idaho Department of Water Resources | 2010 | Irrigation (Watermaster) | http://www.idwr.idaho.gov/ftp/gisdata/Spatial/WaterRights |






Table 2 Crop categories used to approximate the land use categories in the ET calculation

| Land use category | Approximated Crops in ET calculation |
| --- | --- |
| Agricultural | Alfalfa |
| Developed land | Bare land |
| Forest | Poplar |
| Shrubland | Sagebrush |
| Herbaceous | Average of Cheatgrass, bunch grass and bromegrass |




Table 3 Parameters used, the range considered for calibration and the calibrated values

| Routine | Parameter | Description | Units | Range Considered | Calibrated Value |
|---|---|---|---|---|---|
| **Snow Routine** | TT | Threshold temperature | ºC | -2.0 - 2.0 | 0.4 |
| | CFMAX | Degree-day factor governing maximum snowmelt rate | mm/ºC /day | 1.0 - 6.0 | 3.6 |
| | SFCF | Snowmelt correction factor | - | 0.5 - 3.0 | 2.2 |
| | CFR | Refreeze coefficient | - | 0.05 | 0.05 |
| | CWH | Water holding capacity of snowpack | - | 0.1 | 0.1 |
| **Soil and Evaporation Routine** | FC | Maximum depth of water in soil water reservoir | mm | 395 | 395 |
| | LP | Soil moisture value above which actual ET = PET | mm | 200.0 | 200 |
| | WP | Wilting point in soil for ET to occur | mm | 147 | 147 |
| | BETA | Shaping Coefficient | - | 1.0 - 6.0 | 2.6 |
| **Ground-water and Response Routine** | PERC | Percolation coefficient | per day | 0.1 - 10.0 | 6.6 |
| | UZL | Threshold for K0 to outflow | mm | 10.0 - 500.0 | 240.7 |
| | K0 | Recession coefficient | per day | 0.1 - 1 | 0.7 |
| | K1 | Recession coefficient | per day | 0.01 - 1.0 | 0.07 |
| | K2 | Recession coefficient | per day | 0.0001 - 1.0 | 0.0002 |







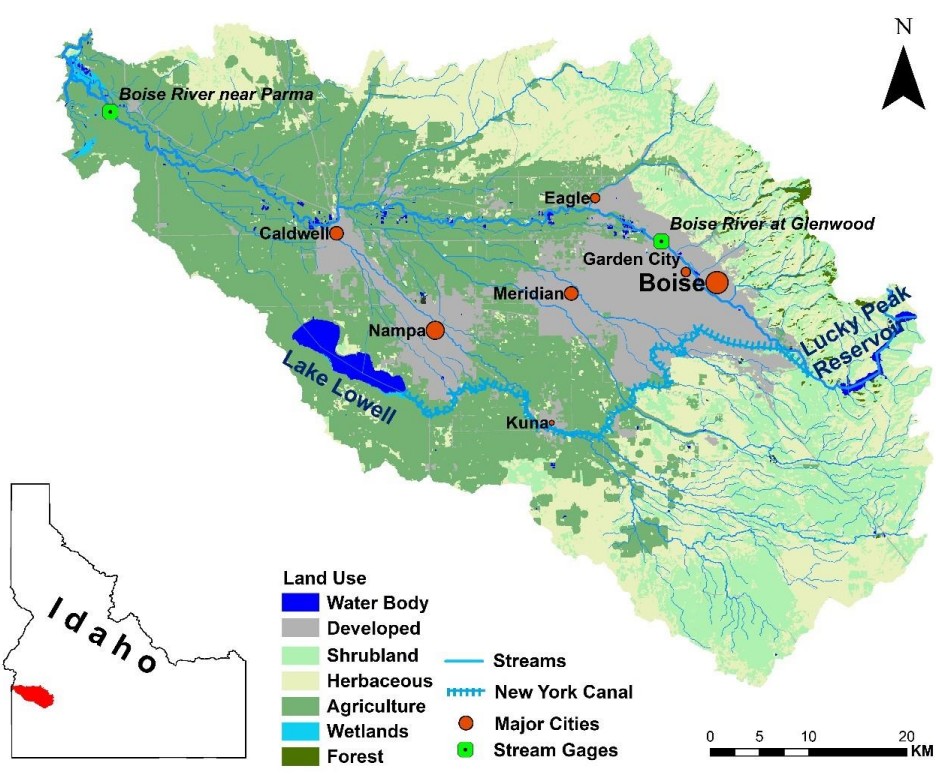


Figure 1 Study Area: The Treasure Valley which is located at Southwest Idaho, with Idaho's
three largest cities and complex agricultural activities.






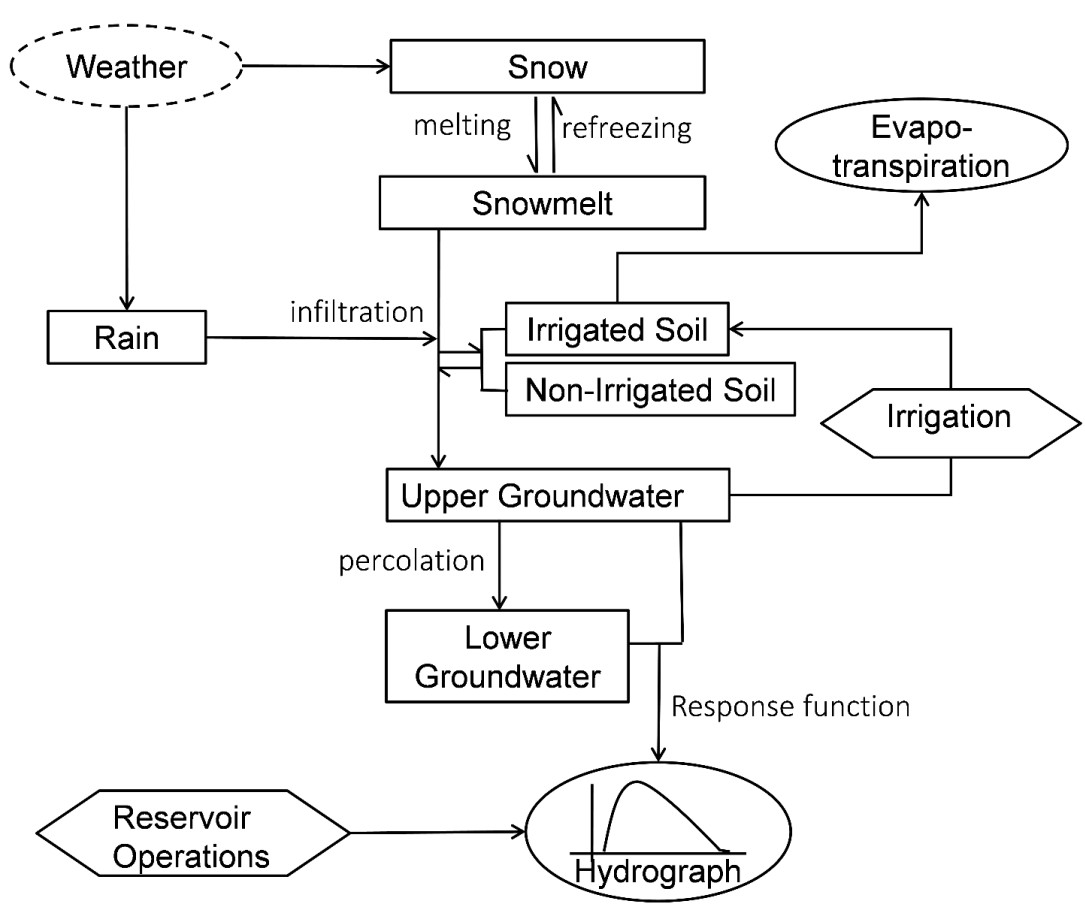


Figure 2 Flowchart of the Flow model in Envision. Note the human activities influencing water
availability. Water is distributed by the local water rights data (irrigation activities), and is also
constrained by the reservoir operations.





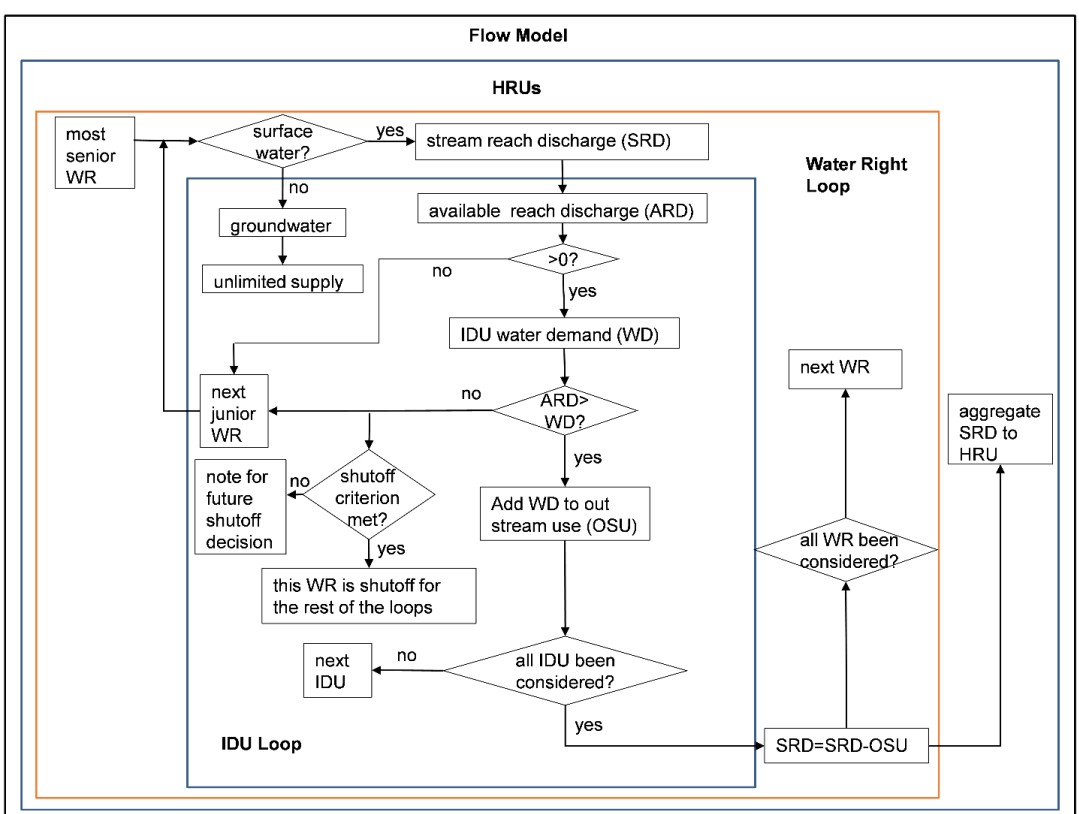

Figure 3 Flowchart of the water right loop in Envision. Each water right is first appropriated for
each IDU it applies to. At each flow time step, the stream reach discharge is then aggregated to
the HRU level, and used for the next time step.





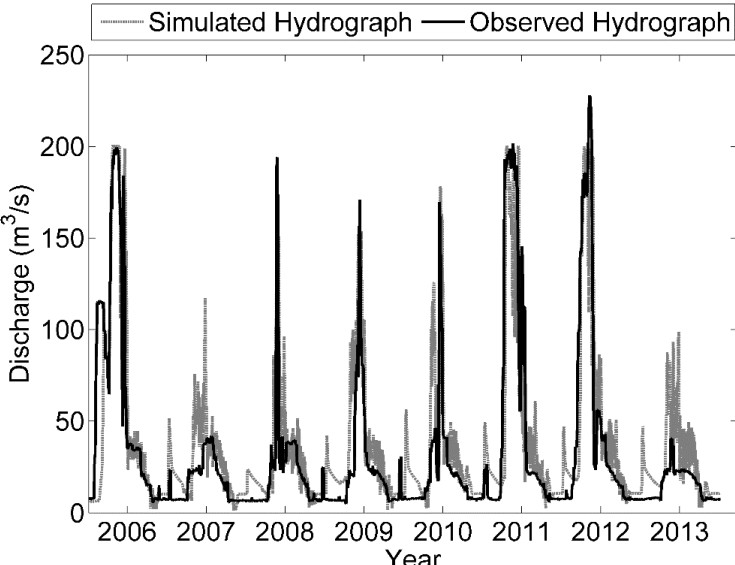


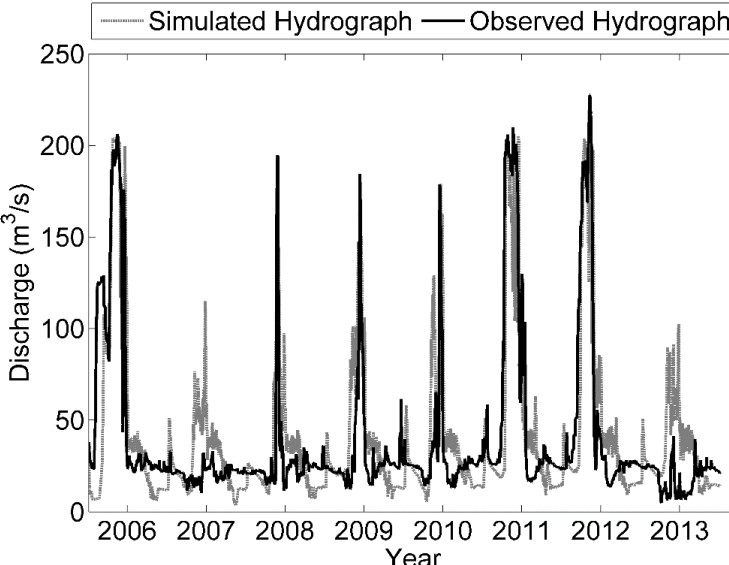



Figure 4 Simulated discharge and the observations during the calibration (2006 ~ 2009) and
validation periods (2010 ~ 2013) at the Glenwood Station of Boise River (Upper Panel) and
Parma Station of Boise River (Lower Panel).






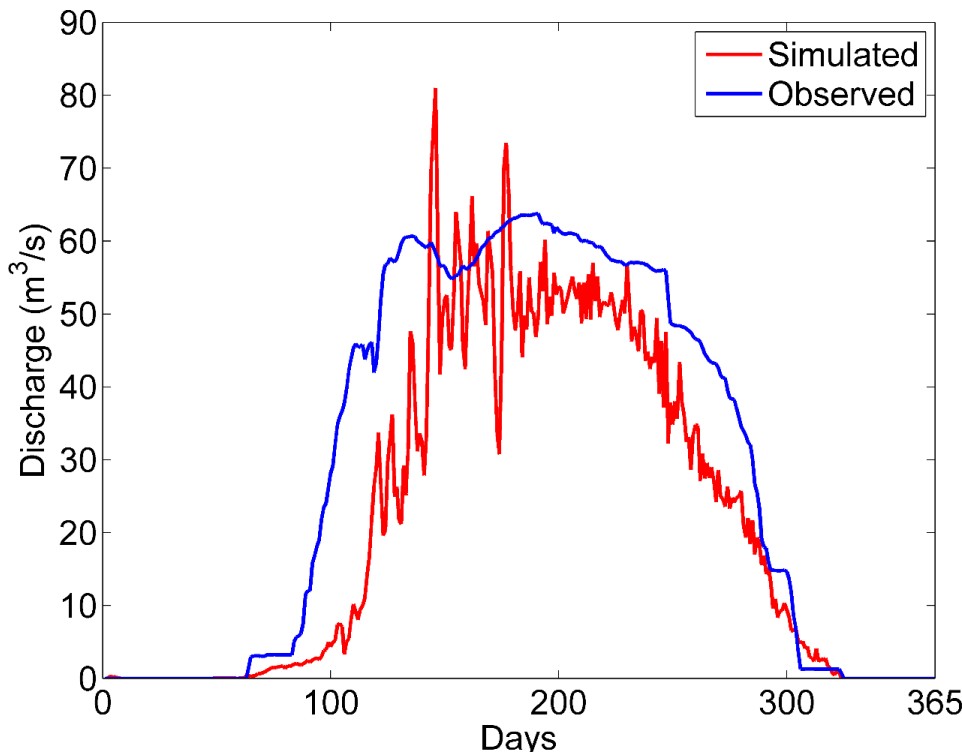


Figure 5 Simulated irrigation amount and the observations averaged over the years of 2006 ~
2013 at the New York Canal. Blue color lines are daily discharge rate in m^3/s, and red color
lines are cumulative discharge in m^3.





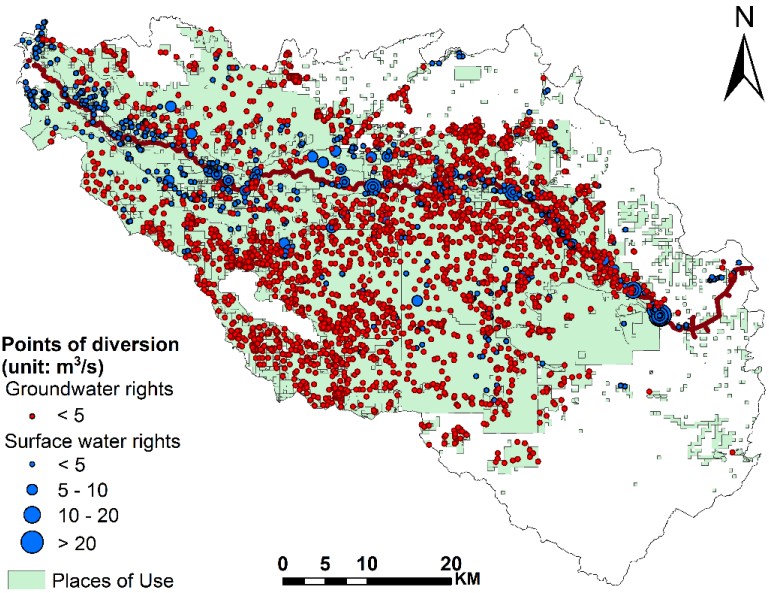

Figure 6 The maximum allowed diversion rates and the spatial distribution of the Points of
Diversion (PODs). Note that multiple diversion PODs overlap at the New York Canal diversion
places, and the water diverted from New York Canal serves as the main surface water
resources for the agricultural areas.






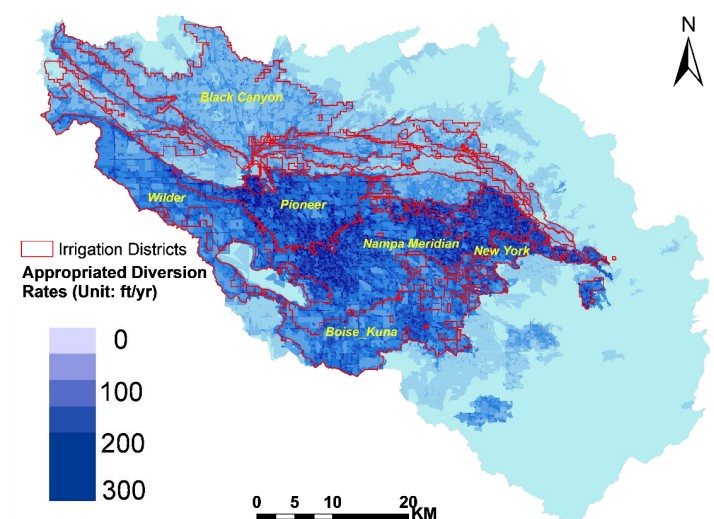

Figure 7 The annual appropriated diversion rates calculated based on water rights maximum
allowed diversion rates and place of use, indicating the potential usable water. The irrigation
district boundaries and the names of major irrigation districts are also shown.





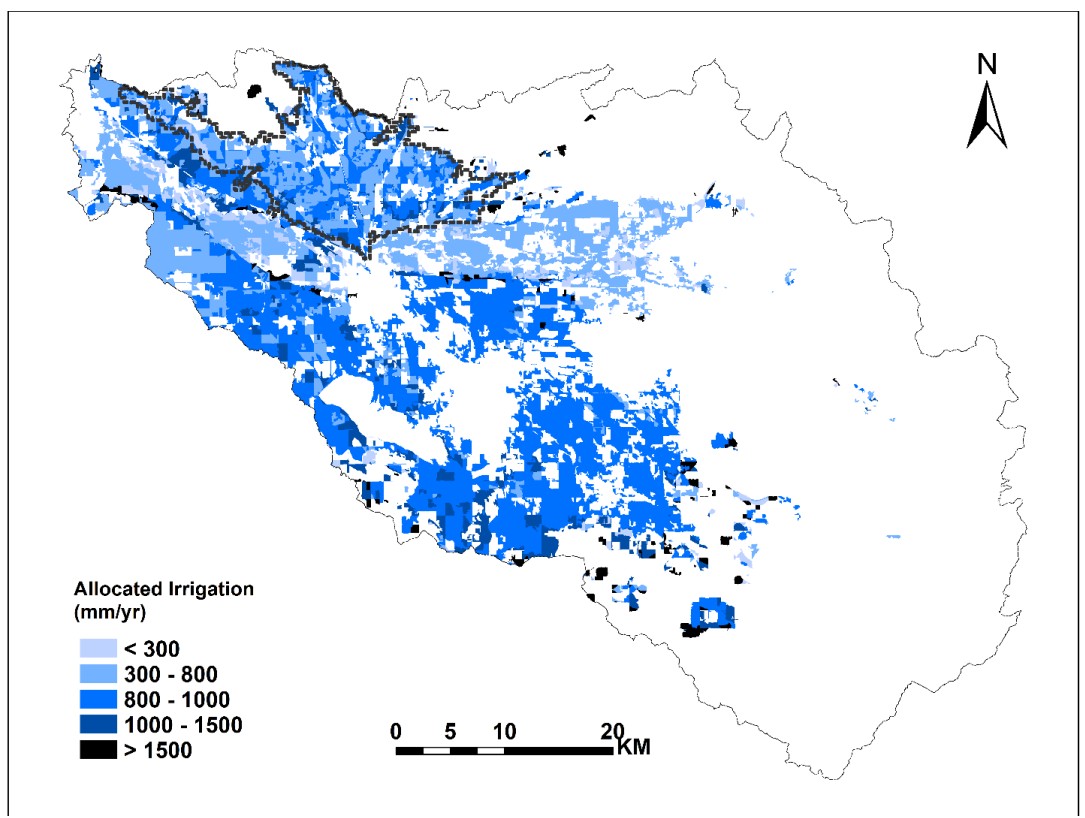

Figure 8 The spatial distribution of the annual allocated irrigation water averaged over the
simulation period. The domain that is within the dotted circle is Black Canyon Irrigation District,
which receives additional irrigation water from outside of the domain, where the water allocation
is underestimated.




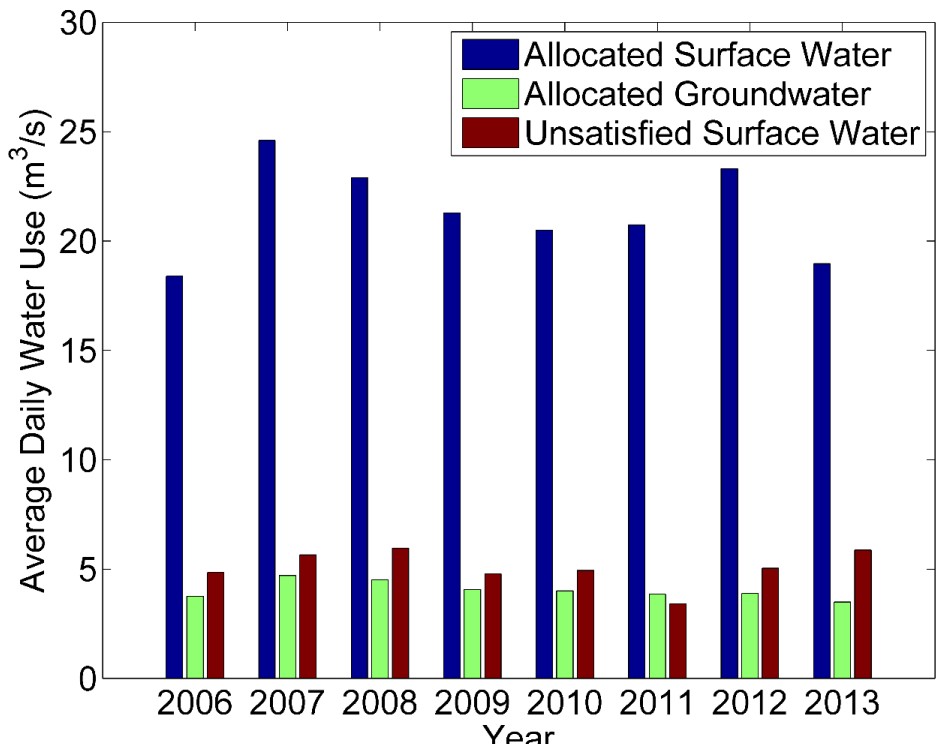

Figure 9: Average daily allocated surface water, groundwater and unsatisfied surface water use
for each year.





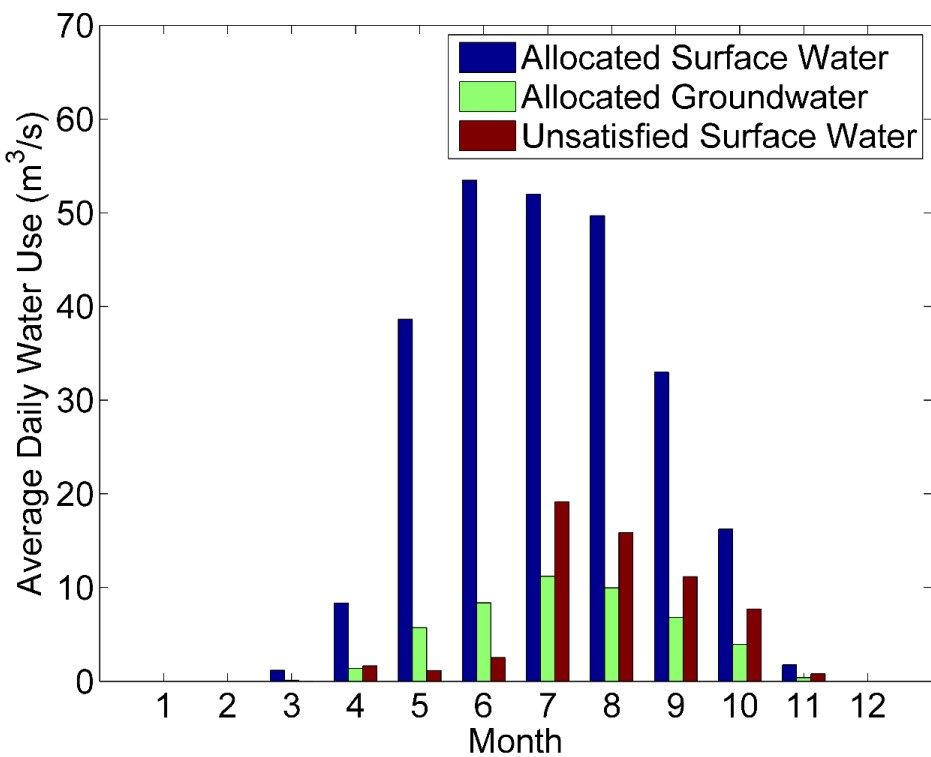

Figure 10: Average daily allocated surface water, groundwater and unsatisfied surface water use
for each month from 2006 to 2013.





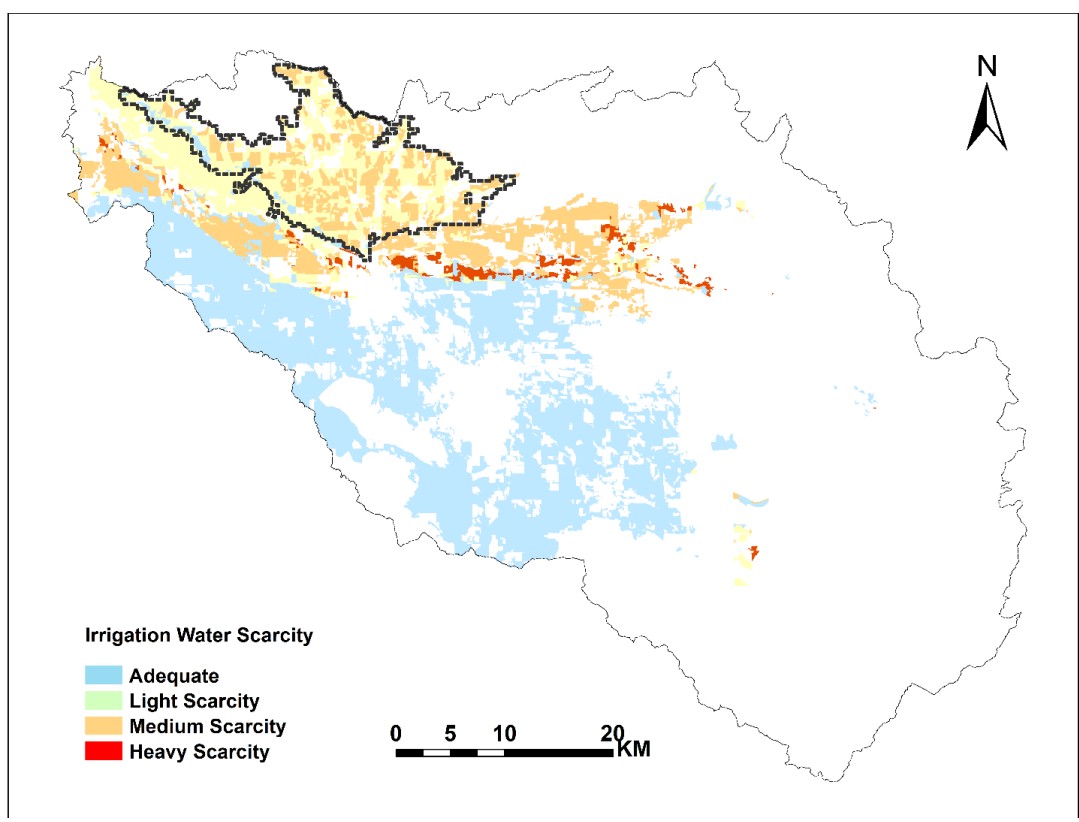

Figure 11: The spatial distribution of the annual unsatisfied irrigation maps averaged over the
simulation period. The domain that is circled is Black Canyon Irrigation District, which receives
additional irrigation water from outside of the domain, where the water scarcity is overestimated.



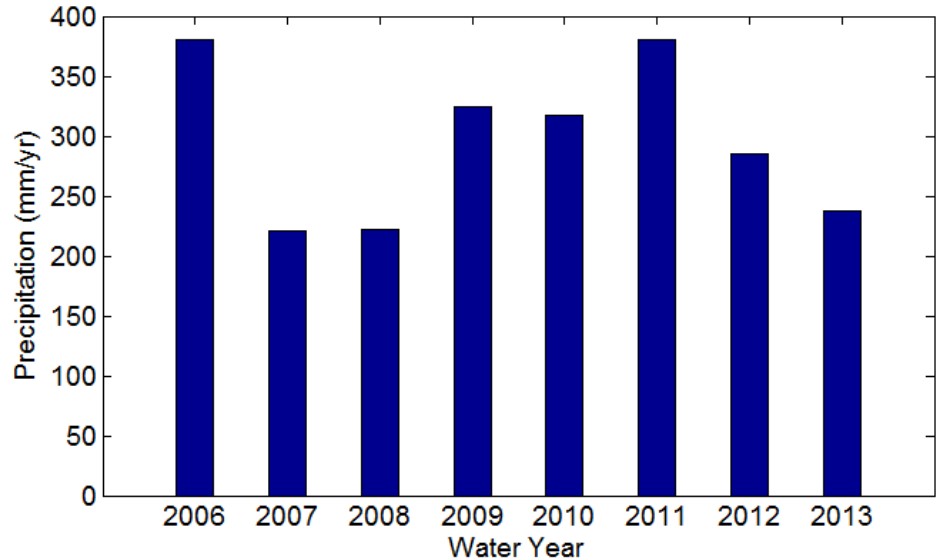

Figure 12 Annual precipitation amount calculated at Boise Air Terminal (Station ID:
7268104131). Precipitation is calculated based on water year since irrigation in each calenda

810          year is mainly affected by the precipitation during the spring and last winter.






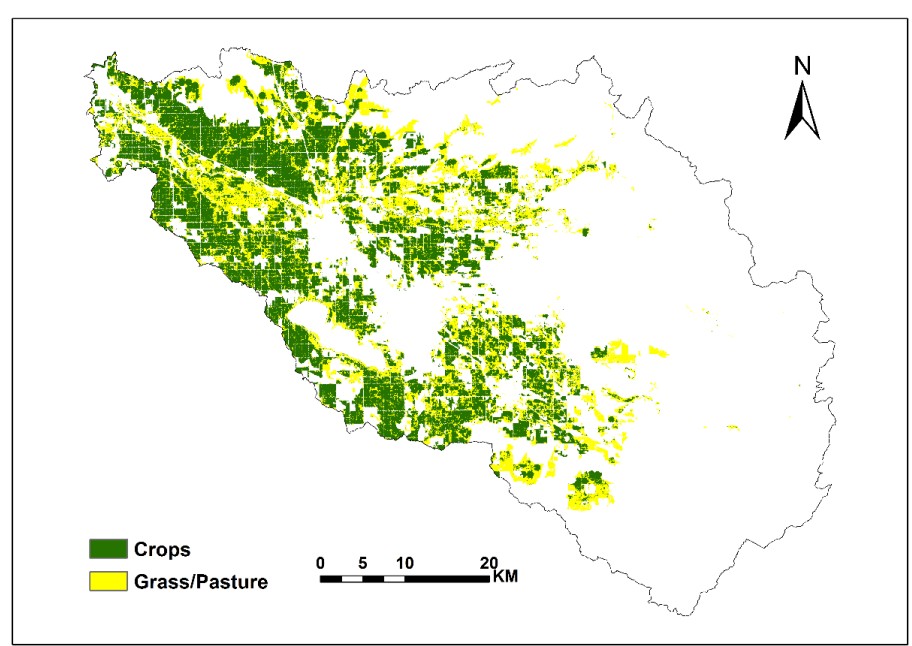


Figure 13 The spatial distribution of crops and grass/pasture in the agricultural area of the

Treasure Valley.
