# Peer review of "Coupling biophysical processes and water rights to simulate spatially distributed"

_Hydrology and Earth System Sciences, 2017_

## Referee Comment (RC1) · Anonymous Referee #1 · 23 Mar 2017

Water resources management is a comprehensive issue which integrates hydrology cycle and water use. Water scarcity is a pressing problem in the western U.S., and the management is fairly complex. A solid and easy-to-use modeling tool that incorporates water rights information within a hydrological model will definitely help with water management decision making. However, due to the complexity of simulating the water allocation based on prior appropriation water right doctrine, hydrologic modeling research often does not explicitly include water rights. The few examples include the Water Rights Analysis Package (WRAP) and VIC-Cropsys. The current study presented valuable attempts to simulate the water allocation based on water rights. The study is unique than others in that it not only considers evapotranspiration and soil

moisture in agricultural land, but also considers the water availability simulated in the streams and maximum allowed water quote/seniority based on the water right regulations. As such, it is truly an integration of social and biophysical processes in a modeling framework, considering hydrology, agriculture, and irrigation based on water law. The authors tested their model in the Treasure Valley, Idaho, which is a typical western semi-arid region, and the model is proved to capture the spatial allocation and timing of irrigation water use quite well. The calibration and validation processes seem a bit simple considering the model parameters involved and the complexity of the model. It is, thought, solid. The model is potentially a great tool that is applicable to many places in the western U.S. facing similar water resources challenges and following Prior Appropriation Doctrine. The approach also has the potential to be extended to simulate other water uses (industrial, domestic, municipal, commercial water use etc.) as long as the same prior appropriation doctrine is used. The manuscript is well organized and easy to follow, and the topic is of interest to HESS readers. I would recommend publication of the work with minor revisions. The minor concerns are as follows. 1. The authors calibrated 9 parameters and left 5 parameters as constant. It should be justified how the 5 constant parameters are selected? Based on sensitivity analysis or literature? 2. Figure 5 caption does not match with the content of the figure. 3. Table 1 should list the temporal and/or spatial resolution of the data used in the study. 4. For hydrologic modeling, the longer period of records is always better. However, to model water use, dry years play much bigger role as it is the time competing users need harvest water from hydrologic system simultaneously. The model is calibrated (verified) using 2006 to 2013. Please add short description of the period of records. It is dry or wet when much longer periods are consider? From the figures, one can only see that high flow vary significantly but low flow seems stable over the period. 5. From water resources management perspective, decision making often prefers conservative estimates. If a model is meant to be used to manage water during drought, underestimation of water availability is often preferred than overestimation. I am glad that the authors acknowledged that the limitation of overestimation and provided insights on

possible reasons.

---

## Referee Comment (RC2) · Anonymous Referee #2 · 31 Mar 2017

Han and colleagues address the important challenge of agricultural water management
in a region prone to water stress. They develop a spatially explicit model of the Trea-
sure Valley area in Idaho, U.S. that couples biophysical processes and water rights.
Specifically, this model aims to diagnose the times and places where water supplies
are insufficient to meet agricultural demands by incorporating the quantity and senior-
ity of water rights from the Boise River. Irrigation water significantly alters the water
balance and its application is determined not just by hydrological availability but the
laws governing water rights. The integration of water rights in a spatially explicit model
has the potential to lead to new insights on the challenges of water management and
the opportunities for improvement. The manuscript is well written and the topic is of

interest to Hydrology and Earth Systems Science readers. However, I do have a series of minor comments that would strengthen the paper. I recommend publication after minor revisions.

1) The terms defined starting on line 315 would be clearer in a numbered or bulleted list.

2) On line 339 'simulates' should read 'simulating.'

3) The Nash-Sutcliffe Efficiency Coefficient is referred to as both the 'Nash-Sutcliffe Coefficient' (line 352) and the 'Nash-Sutcliffe Efficiency' (line 366) and abbreviated as both 'NS' (line 399) and 'E' (line 366). Please revise for consistency.

4) In Figure 4, label the two panels a and b or similar for clarity

5) In the model, the reservoir operations pass through natural flows within target range. However, fall flows at the Parma Station are consistently under predicted. Please discuss the potential causes of this discrepancy.

6) Figure 5 is hard to read in black and white. Making this figure consistent with Figure 4 would resolve the issue.

7) Figures 7 and 8 offer a useful visual to compare the spatial allocation of water based on water rights and the modeled spatial allocation of water. However, the different units (feet vs. mm) make this comparison misleading. Please revise using consistent units, color scheme, and scale.

8) In Figures 8 and 11 the domain is circled not outlined as noted in the caption. Please revise for clarity.

9) On line 455 note the average surface and groundwater usage in the model and Figure 10 shows the average unsatisfied surface water per month. Is there any available data to compare these results to? Are summer water shortages reported by local farmers?

[Figure]

10) How does Figure 9 support the claim that allocated water is a complex nonlinear issue (line 553)?

11) On line 566 'corporation' should read 'cooperation.'

12) This model assumes all farmers make irrigation decisions rationally based on water availability. However, the heterogeneity of decision making may have important implications here (see Noel and Cai 2017). I understand that an analysis of this is out of the scope of the current work, but speaking to the implications of rational decision making as a simplifying assumption would augment the discussion section.

References

Noël, P. H., & Cai, X. (2017). On the role of individuals in models of coupled human and natural systems: Lessons from a case study in the Republican River Basin. Environmental Modelling & Software, 92(March 1993), 1–16. http://doi.org/10.1016/j.envsoft.2017.02.010

---

## Referee Comment (RC3) · Anonymous Referee #3 · 3 Apr 2017

The paper "Coupling biophysical processes and water rights to simulate spatially distributed water use in an intensively managed hydrologic system" by Han et al. presents a modelling framework to integrate water rights allocation into a hydrologic model capture the spatial distribution of irrigation water diversion in semi-arid basins in Western US. Agricultural irrigation is the largest water consumption, but the socioeconomic and institutional factors affecting irrigation behavior are generally not well represented in hydrologic models. This paper provides an effort to better representing anthropogenic factors in biophysical models and will provide insights on how better water use regulation will support sustainability of water resources management. The paper is well-written and the results are clearly presented. I would suggest a minor revision to the

manuscript. Below are some specific comments: In Line 292, how is water diversion water loss handled in the model? Is diversion water loss added to soil or groundwater or river near the diversion channel? Speaking of irrigation return flow, will the water loss be considered as return flow? Due to the significant amount of water loss (60% of diverted water), more details are needed. This would also provide important information about how irrigation efficiency will affect water allocation and stream flow. In Line 190, the land use and land cover in 2011 is used for the whole simulation. Does the irrigated crop area vary significantly during the simulation period? In Line 294 - Line 306, the irrigation requirements are satisfied based on the seniority of water rights. It would be interesting to see the model results on the allocated or unsatisfied water from different water rights seniority groups. For example, how much water is demanded and actually diverted for different water rights seniority groups? Will senior and junior water rights holders will be affected in wet/dry years? Since the model is unique in representing the water rights, how water is actually diverted to different water right seniority groups would provide important information for water resources management. The unit of y axis in Figure 5 is misleading. The blue color is for discharge rate ($m^3/s$), while the red line is discharge volume ($m^3$). Is it possible to represent the simulated and observed irrigation water in a same unit? The black dash line of Black Canyon Irrigation District in Figure 8 is difficult to capture. In addition, the average annual allocated irrigation water is some places are more than 1000 mm/yr, or even more than 1500 mm/yr. It seems to me the irrigation amount is quite big. Will farmers in these regions apply some much water in the fields? Farmers' irrigation behaviors are affected by many factors, such as irrigation technology, insurance, farmer's preference on profit/risk. Although these are beyond the scope of this study, the authors should briefly discuss it and cite some existing literature on how farmers' behavior affect the hydrologic systems.

---

## Author Comment (AC1) · 3 Apr 2017

Water resources management is a comprehensive issue which integrates hydrology cycle and water use. Water scarcity is a pressing problem in the western U.S., and the management is fairly complex. A solid and easy-to-use modeling tool that incorporates water rights information within a hydrological model will definitely help with water management decision making. However, due to the complexity of simulating the water allocation based on prior appropriation water right doctrine, hydrologic modeling research often does not explicitly include water rights. The few examples include the Water Rights Analysis Package (WRAP) and VIC-Cropsys. The current study presented valuable attempts to simulate the water allocation based on water rights. The study is unique than others in that it not only considers evapotranspiration and soil moisture in agricultural land, but also considers the water availability simulated in the streams and maximum allowed water quote/seniority based on the water right regulations. As such, it is truly an integration of social and biophysical processes in a modeling framework, considering hydrology, agriculture, and irrigation based on water law. The authors tested their model in the Treasure Valley, Idaho, which is a typical western semi-arid region, and the model is proved to capture the spatial allocation and timing of irrigation water use quite well. The calibration and validation processes seem a bit simple considering the model parameters involved and the complexity of the model. It is, thought, solid. The model is potentially a great tool that is applicable to many places in the western U.S. facing similar water resources challenges and following Prior Appropriation Doctrine. The approach also has the potential to be extended to simulate other water uses (industrial, domestic, municipal, commercial water use etc.) as long as the same prior appropriation doctrine is used. The manuscript is well organized and easy to follow, and the topic is of interest to HESS readers. I would recommend publication of the work with minor revisions.

Response: We are very glad that the reviewer agrees with our contribution. We appreciate the reviewer nicely summarized the key aspects of our study and pointed out their importance. Below, we respond to the minor concerns from the reviewer in details.

The minor concerns are as follows.

1. The authors calibrated 9 parameters and left 5 parameters as constant. It should be justified how the 5 constant parameters are selected? Based on sensitivity analysis or literature?

   Response: The selection of parameters for calibration is based on a combination of literature review and data availability. The reasons can be summarized as follows:

   (1) HBV is not a new model itself, and has a rich literature to guide on parameter selection. It has been widely tested that LP, CFR and CWH are not sensitive to model performance [Seibert, 1997]. We will include more references in the revision.

   (2) FC and WP values are readily available data for the region. The watershed is quite small and has relatively uniform soil characteristics in agricultural regions, which can be reflected from the NRCS data sources. We will include the citation link of the values in the revision.

2. Figure 5 caption does not match with the content of the figure.

Response: Thank you for the sharp catch. We revised the figure to remove cumulative values which made the daily comparison less clear. We will definitely change the title back in the revision.

3. Table 1 should list the temporal and/or spatial resolution of the data used in the study.

Response: Thank you for the suggestion. We will include temporal and/or spatial resolution for all the datasets that are applicable.

4. For hydrologic modeling, the longer period of records is always better. However, to model water use, dry years play much bigger role as it is the time competing users need harvest water from hydrologic system simultaneously. The model is calibrated (verified) using 2006 to 2013. Please add short description of the period of records. It is dry or wet when much longer periods are consider? From the figures, one can only see that high flow vary significantly but low flow seems stable over the period.

Response: Thank you for the suggestion. We will include temporal and/or spatial resolution for all the datasets that are applicable in our revision.

5. From water resources management perspective, decision making often prefers conservative estimates. If a model is meant to be used to manage water during drought, underestimation of water availability is often preferred than overestimation. I am glad that the authors acknowledged that the limitation of overestimation and provided insights on possible reasons.

Response: Thank you for the comment.

---

## Author Comment (AC2) · 4 Apr 2017

Water resources management is a comprehensive issue which integrates hydrology cycle and water use. Water scarcity is a pressing problem in the western U.S., and the management is fairly complex. A solid and easy-to-use modeling tool that incorporates water rights information within a hydrological model will definitely help with water management decision making. However, due to the complexity of simulating the water allocation based on prior appropriation water right doctrine, hydrologic modeling research often does not explicitly include water rights. The few examples include the Water Rights Analysis Package (WRAP) and VIC-Cropsys. The current study presented valuable attempts to simulate the water allocation based on water rights. The study is unique than others in that it not only considers evapotranspiration and soil moisture in agricultural land, but also considers the water availability simulated in the streams and maximum allowed water quote/seniority based on the water right regulations. As such, it is truly an integration of social and biophysical processes in a modeling framework, considering hydrology, agriculture, and irrigation based on water law. The authors tested their model in the Treasure Valley, Idaho, which is a typical western semi-arid region, and the model is proved to capture the spatial allocation and timing of irrigation water use quite well. The calibration and validation processes seem a bit simple considering the model parameters involved and the complexity of the model. It is, thought, solid. The model is potentially a great tool that is applicable to many places in the western U.S. facing similar water resources challenges and following Prior Appropriation Doctrine. The approach also has the potential to be extended to simulate other water uses (industrial, domestic, municipal, commercial water use etc.) as long as the same prior appropriation doctrine is used. The manuscript is well organized and easy to follow, and the topic is of interest to HESS readers. I would recommend publication of the work with minor revisions.

Response: We are very glad that the reviewer agrees with our contribution. We appreciate the reviewer nicely summarized the key aspects of our study and pointed out their importance. Below, we respond to the minor concerns from the reviewer in details.

The minor concerns are as follows.

1. The authors calibrated 9 parameters and left 5 parameters as constant. It should be justified how the 5 constant parameters are selected? Based on sensitivity analysis or literature?

   Response: The selection of parameters for calibration is based on a combination of literature review and data availability. The reasons can be summarized as follows:

   (1) HBV is not a new model itself, and has a rich literature to guide on parameter selection. It has been widely tested that LP, CFR and CWH are not sensitive to model performance [Seibert, 1997]. We will include more references in the revision.

   (2) FC and WP values are readily available data for the region. The watershed is quite small and has relatively uniform soil characteristics in agricultural regions, which can be reflected from the NRCS data sources. We will include the citation link of the values in the revision.

2. Figure 5 caption does not match with the content of the figure.

Response: Thank you for the sharp catch. We revised the figure to remove cumulative values which made the daily comparison less clear. We will definitely change the title back in the revision.

3. Table 1 should list the temporal and/or spatial resolution of the data used in the study.

Response: Thank you for the suggestion. We will include temporal and/or spatial resolution for all the datasets that are applicable.

4. For hydrologic modeling, the longer period of records is always better. However, to model water use, dry years play much bigger role as it is the time competing users need harvest water from hydrologic system simultaneously. The model is calibrated (verified) using 2006 to 2013. Please add short description of the period of records. It is dry or wet when much longer periods are consider? From the figures, one can only see that high flow vary significantly but low flow seems stable over the period.

Response: Thank you for the suggestion. This is a very critical point. The reviewer is definitely correct that longer calibration and validation period will be better.

The water use in the Treasure Valley is quite unique in that the water released from the upstream reservoir controls the water amount for downstream users as shown in the hydrograph. From the hydrograph, one can easily see a similar water use pattern every year starting from late spring. The reason is that the upstream reservoir is used for both flood control and irrigation, and has limited storage. If the snow accumulation is high, in early spring, water needs to be released to make sure that downstream city is safe, which reflects the high discharge in the "wet" years. From 2006 to 2013, we have typical wet years (2006, 2008, 2011, 2012) and dry years (2007 and 2013) included, and the relatively shorter periods saves a lot of computational time. So, we used the periods of 2006 ~ 2013 for calibration and validation purposes.

We will add some sentence describing the reasons for the selection of the period of records.

5. From water resources management perspective, decision making often prefers conservative estimates. If a model is meant to be used to manage water during drought, underestimation of water availability is often preferred than overestimation. I am glad that the authors acknowledged that the limitation of overestimation and provided insights on possible reasons.

Response: Thank you for the comment.

---

## Author Comment (AC3) · 14 Apr 2017

**Response to Anonymous Referee #2**

Han and colleagues address the important challenge of agricultural water management in a region prone to water stress. They develop a spatially explicit model of the Treasure Valley area in Idaho, U.S. that couples biophysical processes and water rights. Specifically, this model aims to diagnose the times and places where water supplies are insufficient to meet agricultural demands by incorporating the quantity and seniority of water rights from the Boise River. Irrigation water significantly alters the water balance and its application is determined not just by hydrological availability but the laws governing water rights. The integration of water rights in a spatially explicit model has the potential to lead to new insights on the challenges of water management and the opportunities for improvement. The manuscript is well written and the topic is of interest to Hydrology and Earth Systems Science readers. However, I do have a series of minor comments that would strengthen the paper. I recommend publication after minor revisions.

Response: We appreciate the reviewer's summarization of our research, and the encouragement on publication. We are addressing the reviewer's comments below in details, and will have all of them included in our revision.

1) The terms defined starting on line 315 would be clearer in a numbered or bulleted list.

Response: We use numbered list in the revision.

2) On line 339 'simulates' should read 'simulating.'

Response: We will change the word in the revision.

3) The Nash-Sutcliffe Efficiency Coefficient is referred to as both the 'Nash-Sutcliffe Coefficient' (line 352) and the 'Nash-Sutcliffe Efficiency' (line 366) and abbreviated as both 'NS' (line 399) and 'E' (line 366). Please revise for consistency.

Response: Thank you for catching the inconsistencies. We will consistently use 'Nash-Sutcliffe Coefficient' and abbreviate it as 'NS' in the revision.

4) In Figure 4, label the two panels a and b or similar for clarity

Response: Thank you for the suggestion. We will label the panels as suggested.

5) In the model, the reservoir operations pass through natural flows within target range. However, fall flows at the Parma Station are consistently under predicted. Please discuss the potential causes of this discrepancy.

Response: We appreciate the reviewer bringing up the point. We realized this issue and discussed about it in lines 386 – 393 and lines 527 - 537. The major reasons are: 1) The model groundwater supply is assumed to be unlimited for the current situation. This reflects the truth of the current situation and simplifies the model, but will lead to unbalance of water budget; 2) The water pumped out of the watershed has not been considered in the current study. This is a relatively small portion of water use, but will specifically affect the discharge at Parma River station. The water management has to deal with the conflicts between political boundaries and watershed boundaries, and that is one of the directions for further work.

6) Figure 5 is hard to read in black and white. Making this figure consistent with Figure 4 would resolve the issue.

Response: We apologize for placing the wrong caption for the figure, and will correct it in the revision. We will also change the line style to make it more readable in black and white.

7) Figures 7 and 8 offer a useful visual to compare the spatial allocation of water based on water rights and the modeled spatial allocation of water. However, the different units (feet vs. mm) make this comparison misleading. Please revise using consistent units, color scheme, and scale.

Response: Thank you for the suggestion. We will consistently use the SI units in the revision.

8) In Figures 8 and 11 the domain is circled not outlined as noted in the caption. Please revise for clarity.

Response: We have had the domain circled in the figure. The confusion may be due to the black and white printing, but we will definitely double check to make sure it is clear in the revision.

9) On line 455 note the average surface and groundwater usage in the model and Figure 10 shows the average unsatisfied surface water per month. Is there any available data to compare these results to? Are summer water shortages reported by local farmers?

Response: Unfortunately, there are no quantified numbers to compare to. The summer shortages have been reported by local farmers through our stakeholder conversations, and that is a big concern for local farmers right now. But so far, we do not have quantified data for that.

10) How does Figure 9 support the claim that allocated water is a complex nonlinear issue (line 553)?

Response: We will reword the sentence to make it clear. We meant to remind readers that the water allocation and water scarcity in a certain year is not linearly related to the current year precipitation amount. Figure 9 can demonstrate that water allocation is high in the dry year 2007 as irrigation water can be received from the snow fall from the previous year.

11) On line 566 'corporation' should read 'cooperation.'

Response: Thank you. We will change it.

12) This model assumes all farmers make irrigation decisions rationally based on water availability. However, the heterogeneity of decision making may have important implications here (see Noel and Cai 2017). I understand that an analysis of this is out of the scope of the current work, but speaking to the implications of rational decision making as a simplifying assumption would augment the discussion section.

Response: Thank you. We will add discussions on the complexity of decision making.

---

## Author Comment (AC4) · 14 Apr 2017

**Response to Anonymous Referee #3**

The paper "Coupling biophysical processes and water rights to simulate spatially distributed water use in an intensively managed hydrologic system" by Han et al. presents a modelling framework to integrate water rights allocation into a hydrologic model capture the spatial distribution of irrigation water diversion in semi-arid basins in Western US. Agricultural irrigation is the largest water consumption, but the socioeconomic and institutional factors affecting irrigation behavior are generally not well represented in hydrologic models. This paper provides an effort to better representing anthropogenic factors in biophysical models and will provide insights on how better water use regulation will support sustainability of water resources management. The paper is well written and the results are clearly presented. I would suggest a minor revision to the manuscript. Below are some specific comments:

Response: We appreciate the positive feedback from the reviewer, and are very happy to respond to the specific questions below.

In Line 292, how is water diversion water loss handled in the model? Is diversion water loss added to soil or groundwater or river near the diversion channel? Speaking of irrigation return flow, will the water loss be considered as return flow? Due to the significant amount of water loss (60% of diverted water), more details are needed. This would also provide important information about how irrigation efficiency will affect water allocation and stream flow.

Response: Water loss is a very complex issue related to seepage along the canals, evapotranspiration, direct flow back to streams etc. The model has no way to capture those details, nor do we have observational data to support the simulation of those water loss details. As such, we took a simple approach by assigning a lump-sum coefficient to reflect the whole water loss. The "lost" water is still applied to the irrigation land, so that it can either evaporates or infiltrates. Part of the infiltrated water will be routed to the stream based on the HBV model. In this way, we are able to capture the diversion rate from the stream correctly. As such, the actual spatial allocation rate in the farm land will be in a smaller scale than the simulation result as part of the water is lost along the canals before arriving at the farmlands. We will include the relevant information in our revision.

In Line 190, the land use and land cover in 2011 is used for the whole simulation. Does the irrigated crop area vary significantly during the simulation period?

Response: Thank you for pointing out this important issue. For this study, we temporarily used the 2011 land use data for the whole simulation. There is certainly land use change over the years, but for the 8-year simulation period, the change is not significant. Our next step is to project the future water use until 2100, and land use change will be a key factor to consider in the long run.

In Line 294 - Line 306, the irrigation requirements are satisfied based on the seniority of water rights. It would be interesting to see the model results on the allocated or unsatisfied water from different water rights seniority groups. For example, how much water is demanded and actually diverted for different water rights seniority groups? Will senior and junior water rights holders will be affected in wet/dry years? Since the model is unique in representing the water rights, how water is actually diverted to different water right seniority groups would provide important information for water resources management.

Response: Thank you. We totally agree. The water rights that are shut off or suspended are very important factors to inform stakeholders. In the current work, we are not able to fully capture those information, but we are trying to have more parameters summarized in our future work.

The unit of y axis in Figure 5 is misleading. The blue color is for discharge rate (mˆ3/s), while the red line is discharge volume (mˆ3). Is it possible to represent the simulated and observed irrigation water in a same unit?

Response: We apologize for this mistake. All the reviewers have pointed out this issue. We will address it in the revision.

The black dash line of Black Canyon Irrigation District in Figure 8 is difficult to capture. In addition, the average annual allocated irrigation water is some places are more than 1000 mm/yr, or even more than 1500 mm/yr. It seems to me the irrigation amount is quite big. Will farmers in these regions apply some much water in the fields?

Response: Thank you. We will change the way how the Black Canyon Irrigation District is reflected in the revision.
With regard to the allocation amount, the number is higher than actual value. The reason is that part of the "loss" water is applied to the farmland. This problem has been answered earlier above.

Farmers' irrigation behaviors are affected by many factors, such as irrigation technology, insurance, farmer's preference on profit/risk. Although these are beyond the scope of this study, the authors should briefly discuss it and cite some existing literature on how farmers' behavior affect the hydrologic systems.

Response: Thank you. We will add discussions on the complexity of farmer's decision making in the revision.